# Reliable training and estimation of variance networks

**Nicki S. Detlefsen**[* †]
nsde@dtu.dk

**Martin Jørgensen**[* †]
marjor@dtu.dk

**Søren Hauberg** [†]
sohau@dtu.dk

## Abstract

We propose and investigate new complementary methodologies for estimating predictive variance networks in regression neural networks. We derive a locally aware mini-batching scheme that results in sparse robust gradients, and we show how to make unbiased weight updates to a variance network. Further, we formulate a heuristic for robustly fitting both the mean and variance networks post hoc. Finally, we take inspiration from posterior Gaussian processes and propose a network architecture with similar extrapolation properties to Gaussian processes. The proposed methodologies are complementary, and improve upon baseline methods individually. Experimentally, we investigate the impact of predictive uncertainty on multiple datasets and tasks ranging from regression, active learning and generative modeling. Experiments consistently show significant improvements in predictive uncertainty estimation over state-of-the-art methods across tasks and datasets.

## 1 Introduction

The quality of *mean* predictions has dramatically increased in the last decade with the rediscovery of neural networks [LeCun et al., 2015]. The predictive *variance*, however, has turned out to be a more elusive target, with established solutions being subpar. The general finding is that neural networks tend to make overconfident predictions [Guo et al., 2017] that can be harmful or offensive [Amodei et al., 2016]. This may be explained by neural networks being general function estimators that does not come with principled uncertainty estimates. Another explanation is that *variance* estimation is a fundamentally different task than *mean* estimation, and that the tools for mean estimation perhaps do not generalize. We focus on the latter hypothesis within regression.

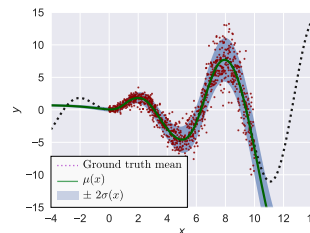

To illustrate the main practical problems in variance estimation, we consider a toy problem where data is generated as $y = x \cdot \sin(x) + 0.3 \cdot \epsilon_1 + 0.3 \cdot x \cdot \epsilon_2$, with $\epsilon_1, \epsilon_2 \sim \mathcal{N}(0, 1)$ and $x$ is uniform on $[0, 10]$ (Fig. 1). As is common, we do maximum likelihood estimation of $\mathcal{N}(\mu(x), \sigma^2(x))$, where $\mu$ and $\sigma^2$ are neural nets. While $\mu$ provides an almost perfect fit to the ground truth, $\sigma^2$ shows two problems: $\sigma^2$ is significantly underestimated and $\sigma^2$ does not increase outside the data support to capture the poor mean predictions.

Figure 1: Max. likelihood fit of $\mathcal{N}(\mu(x), \sigma^2(x))$ to data.

These findings are general (Sec. 4), and alleviating them is the main purpose of the present paper. We find that this can be achieved by a combination of methods that *1)* change the usual mini-batching to be location aware; *2)* only optimize variance conditioned on the mean; *3)* for scarce data, we introduce a more robust likelihood function; and *4)* enforce well-behaved interpolation and extrapolation of variances. Points 1 and 2 are achieved through changes to the training algorithm, while 3 and 4 are changes to model specifications. We empirically demonstrate that these new tools significantly improve on state-of-the-art across datasets in tasks ranging from regression to active learning, and generative modeling.

---

[*]Equal contribution
[†]Section for Cognitive Systems, Technical University of Denmark

## 2   Related work

**Gaussian processes (GPs)** are well-known function approximators with built-in uncertainty estimators [Rasmussen and Williams, 2006]. GPs are robust in settings with a low amount of data, and can model a rich class of functions with few hyperparameters. However, GPs are computationally intractable for large amounts of data and limited by the expressiveness of a chosen kernel. Advances like sparse and deep GPs [Snelson and Ghahramani, 2006, Damianou and Lawrence, 2013] partially alleviate this, but neural nets still tend to have more accurate mean predictions.

**Uncertainty aware neural networks** model the predictive mean and variance as two separate neural networks, often as multi-layer perceptrons. This originates with the work of Nix and Weigend [1994] and Bishop [1994]; today, the approach is commonly used for making variational approximations [Kingma and Welling, 2013, Rezende et al., 2014], and it is this general approach we investigate.

**Bayesian neural networks (BNN)** [MacKay, 1992] assume a prior distribution over the network parameters, and approximate the posterior distribution. This gives direct access to the approximate predictive uncertainty. In practice, placing an informative prior over the parameters is non-trivial. Even with advances in stochastic variational inference [Kingma and Welling, 2013, Rezende et al., 2014, Hoffman et al., 2013] and expectation propagation [Hernández-Lobato and Adams, 2015], it is still challenging to perform inference in BNNs.

**Ensemble methods** represent the current state-of-the-art. *Monte Carlo (MC) Dropout* [Gal and Ghahramani, 2016] measure the uncertainty induced by Dropout layers [Hinton et al., 2012] arguing that this is a good proxy for predictive uncertainty. *Deep Ensembles* [Lakshminarayanan et al., 2017] form an ensemble from multiple neural networks trained with different initializations. Both approaches obtain ensembles of *correlated* networks, and the extent to which this biases the predictive uncertainty is unclear. Alternatives include estimating *confidence intervals* instead of variances [Pearce et al., 2018], and gradient-based Bayesian model averaging [Maddox et al., 2019].

**Applications of uncertainty** include *reinforcement learning*, *active learning*, and *Bayesian optimization* [Szepesvári, 2010, Huang et al., 2010, Frazier, 2018]. Here, uncertainty is the crucial element that allows for systematically making a trade-off between *exploration* and *exploitation*. It has also been shown that uncertainty is required to learn the topology of data manifolds [Hauberg, 2018].

**The main categories of uncertainty** are *epistemic* and *aleatoric* uncertainty [Kiureghian and Ditlevsen, 2009, Kendall and Gal, 2017]. Aleatoric uncertainty is induced by unknown or unmeasured features, and, hence, does not vanish in the limit of infinite data. Epistemic uncertainty is often referred to as *model uncertainty*, as it is the uncertainty due to model limitations. It is this type of uncertainty that Bayesian and ensemble methods generally estimate. We focus on the overall *predictive uncertainty*, which reflects both epistemic and aleatoric uncertainty.

## 3   Methods

The opening remarks (Sec. 1) highlighted two common problems that appear when $\mu$ and $\sigma^2$ are neural networks. In this section we analyze these problems and propose solutions.

**Preliminaries.**   We assume that datasets $\mathcal{D} = \{\boldsymbol{x}_i, y_i\}_{i=1}^{N}$ contain i.i.d. observations $y_i \in \mathbb{R}, \boldsymbol{x}_i \in \mathbb{R}^D$. The targets $y_i$ are assumed to be conditionally Gaussian, $p_\theta(y|\boldsymbol{x}) = \mathcal{N}(y|\mu(\boldsymbol{x}), \sigma^2(\boldsymbol{x}))$, where $\mu$ and $\sigma^2$ are continuous functions parametrized by $\theta = \{\theta_\mu, \theta_{\sigma^2}\}$. The maximum likelihood estimate (MLE) of the variance of i.i.d. observations $\{y_i\}_{i=1}^{N}$ is $\frac{1}{N-1}\sum_i (y_i - \hat{\mu})^2$, where $\hat{\mu}$ is the sample mean. This MLE does not exist based on a single observation, unless the mean $\mu$ is known, i.e. the mean is not a free parameter. When $y_i$ is Gaussian, the residuals $(y_i - \mu)^2$ are gamma distributed.

### 3.1   A local likelihood model analysis

By assuming that both $\mu$ and $\sigma^2$ are continuous functions, we are implicitly saying that $\sigma^2(\boldsymbol{x})$ is correlated with $\sigma^2(\boldsymbol{x} + \delta)$ for sufficiently small $\delta$, and similar for $\mu$. Consider the local likelihood estimation problem [Loader, 1999, Tibshirani and Hastie, 1987] at a point $\boldsymbol{x}_i$,

$$\log \tilde{p}_\theta(y_i|\boldsymbol{x}_i) = \sum_{j=1}^{N} w_j(\boldsymbol{x}_i) \log p_\theta(y_j|\boldsymbol{x}_j), \tag{1}$$

where $w_j$ is a function that declines as $\|\boldsymbol{x}_j - \boldsymbol{x}_i\|$ increases, implying that the local likelihood at $\boldsymbol{x}_i$ is dependent on the points nearest to $\boldsymbol{x}_i$. Notice $\tilde{p}_\theta(y_i|\boldsymbol{x}_i) = p_\theta(y_i|\boldsymbol{x}_i)$ if $w_j(\boldsymbol{x}_i) = \mathbf{1}_{i=j}$. Consider, with this $w$, a uniformly drawn subsample (i.e. a standard mini-batch) of the data $\{\boldsymbol{x}_k\}_{k=1}^M$ and its corresponding stochastic gradient of Eq. 1 with respect to $\theta_{\sigma^2}$. If for a point, $\boldsymbol{x}_i$, no points near it are in the subsample, then no other point will influence the gradient of $\sigma^2(\boldsymbol{x}_i)$, which will point in the direction of the MLE, that is highly uninformative as it does not exist unless $\mu(\boldsymbol{x}_i)$ is known. Local data scarcity, thus, implies that while we have sufficient data for fitting a *mean*, locally we have insufficient data for fitting a *variance*. Essentially, if a point is isolated in a mini-batch, all information it carries goes to updating $\mu$ and none is present for $\sigma^2$.

If we do not use mini-batches, we encounter that gradients wrt. $\theta_\mu$ and $\theta_{\sigma^2}$ will both be scaled with $\frac{1}{2\sigma^2(\boldsymbol{x})}$ meaning that points with small variances effectively have higher learning rates [Nix and Weigend, 1994]. This implies a bias towards low-noise regions of data.

### 3.2 Horvitz-Thompson adjusted stochastic gradients

We will now consider a solution to this problem within the local likelihood framework, which will give us a reliable, but biased, stochastic gradient for the usual (nonlocal) log-likelihood. We will then show how this can be turned into an unbiased estimator.

If we are to add some local information, giving more reliable gradients, we should choose a $w$ in Eq.1 that reflects this. Assume for simplicity that $w_j(\boldsymbol{x}_i) = \mathbf{1}_{\|\boldsymbol{x}_i - \boldsymbol{x}_j\| < d}$ for some $d > 0$. The gradient of $\log \tilde{p}_\theta(y|\boldsymbol{x}_i)$ will then be informative, as more than one observation will contribute to the local variance if $d$ is chosen appropriately. Accordingly, we suggest a practical mini-batching algorithm that samples a random point $\boldsymbol{x}_j$ and we let the mini-batch consist of the $k$ nearest neighbors of $\boldsymbol{x}_j$.[3] In order to allow for more variability in a mini-batch, we suggest sampling $m$ points uniformly, and then sampling $n$ points among the $k$ nearest neighbors of each of the $m$ initially sampled points. Note that this is a more informative sample, as all observations in the sample are likely to influence the same subset of parameters in $\theta$, effectively increasing the degrees of freedom[4], hence the quality of variance estimation. In other words, if the variance network is sufficiently expressive, our Monte Carlo gradients under this sampling scheme are of smaller variation and more sparse. In the supplementary material, we empirically show that this estimator yields significantly more sparse gradients, which results in improved convergence. Pseudo-code of this sampling-scheme, can be found in the supplementary material.

While such a mini-batch would give rise to an informative stochastic gradient, it would not be an unbiased stochastic gradient of the (nonlocal) log-likelihood. This can, however, be adjusted by using the *Horvitz-Thompson (HT)* algorithm [Horvitz and Thompson, 1952], i.e. rescaling the log-likelihood contribution of each sample $\boldsymbol{x}_j$ by its inclusion probability $\pi_j$. With this, an unbiased estimate of the log-likelihood (up to an additive constant) becomes

$$\sum_{i=1}^N \left\{ -\frac{1}{2}\log(\sigma^2(\boldsymbol{x}_i)) - \frac{(y_i - \mu(\boldsymbol{x}_i))^2}{2\sigma^2(\boldsymbol{x}_i)} \right\} \approx \sum_{\boldsymbol{x}_j \in \mathcal{O}} \frac{1}{\pi_j} \left\{ -\frac{1}{2}\log(\sigma^2(\boldsymbol{x}_j)) - \frac{(y_j - \mu(\boldsymbol{x}_j))^2}{2\sigma^2(\boldsymbol{x}_j)} \right\} \quad (2)$$

where $\mathcal{O}$ denotes the mini-batch. With the nearest neighbor mini-batching, the inclusion probabilities can be calculated as follows. The probability that observation $j$ is in the sample is $n/k$ if it is among the $k$ nearest neighbors of one of the initial $m$ points, which are chosen with probability $m/N$, i.e.

$$\pi_j = \frac{m}{N} \sum_{i=1}^N \frac{n}{k} \mathbf{1}_{j \in \mathcal{O}_k(i)}, \quad (3)$$

where $\mathcal{O}_k(i)$ denotes the $k$ nearest neighbors of $\boldsymbol{x}_i$.

**Computational costs** The proposed sampling scheme requires an upfront computational cost of $O(N^2 D)$ before any training can begin. We stress that this is pre-training computation and not

updated during training. The cost is therefore relative small, compared to training a neural network for small to medium size datasets. Additionally, we note that the search algorithm does not have to be precise, and we could therefore take advantage of fast approximate nearest neighbor algorithms [Fu and Cai, 2016].

### 3.3 Mean-variance split training

The most common training strategy is to first optimize $\theta_\mu$ assuming a constant $\sigma^2$, and then proceed to optimize $\theta = \{\theta_\mu, \theta_{\sigma^2}\}$ jointly, i.e. a *warm-up* of $\mu$. As previously noted, the MLE of $\sigma^2$ does not exist when only a single observation is available and $\mu$ is unknown. However, the MLE *does* exist when $\mu$ is known, in which case it is $\hat{\sigma}^2(\boldsymbol{x}_i) = (y_i - \mu(\boldsymbol{x}_i))^2$, assuming that the continuity of $\sigma^2$ is not crucial. This observation suggests that the usual training strategy is substandard as $\sigma^2$ is never optimized assuming $\mu$ is known. This is easily solved: we suggest to never updating $\mu$ and $\sigma^2$ simultaneously, i.e. only optimize $\mu$ conditioned on $\sigma^2$, and vice versa. This reads as sequentially optimizing $p_\theta(y|\theta_\mu)$ and $p_\theta(y|\theta_{\sigma^2})$, as we under these conditional distributions we may think of $\mu$ and $\sigma^2$ as known, respectively. We will refer to this as *mean-variance split training (MV)*.

### 3.4 Estimating distributions of variance

When $\sigma^2(\boldsymbol{x}_i)$ is influenced by few observations, underestimation is still likely due to the left skewness of the gamma distribution of $\hat{\sigma}_i^2 = (y_i - \mu(\boldsymbol{x}_i))^2$. As always, when in a low data regime, it is sensible to be Bayesian about it; hence instead of point estimating $\hat{\sigma}_i^2$ we seek to find a distribution. Note that we are not imposing a prior, we are training the parameters of a Bayesian model. We choose the inverse-Gamma distribution, as this is the conjugate prior of $\sigma^2$ when data is Gaussian. This means $\theta_{\sigma^2} = \{\theta_\alpha, \theta_\beta\}$ where $\alpha, \beta > 0$ are the shape and scale parameters of the inverse-Gamma respectively. So the log-likelihood is now calculated by integrating out $\sigma^2$

$$\log p_\theta(y_i) = \log \int \mathcal{N}(y_i|\mu_i, \sigma_i^2)\mathrm{d}\sigma_i^2 = \log t_{\mu_i, \alpha_i, \beta_i}(y_i), \tag{4}$$

where $\sigma_i^2 \sim \text{INV-GAMMA}(\alpha_i, \beta_i)$ and $\alpha_i = \alpha(\boldsymbol{x}_i), \beta_i = \beta(\boldsymbol{x}_i)$ are modeled as neural networks. Having an inverse-Gamma prior changes the predictive distribution to a located-scaled[5] Student-*t* distribution, parametrized with $\mu, \alpha$ and $\beta$. Further, the *t*-distribution is often used as a replacement of the Gaussian when data is scarce and the true variance is unknown and yields a *robust* regression [Gelman et al., 2014, Lange et al., 1989]. We let $\alpha$ and $\beta$ be neural networks that implicitly determine the degrees of freedom and the scaling of the distribution. Recall the higher the degrees of freedom, the better the Gaussian approximation of the *t*-distribution.

### 3.5 Extrapolation architecture

If we evaluate the local log-likelihood (Eq. 1) at a point $\boldsymbol{x}_0$ far away from all data points, then the weights $w_i(\boldsymbol{x}_0)$ will all be near (or exactly) zero. Consequently, the local log-likelihood is approximately 0 regardless of the observed value $y(\boldsymbol{x}_0)$, which should be interpreted as a large entropy of $y(\boldsymbol{x}_0)$. Since we are working with Gaussian and *t*-distributed variables, we can recreate this behavior by exploiting the fact that entropy is only an increasing function of the variance. We can re-enact this behavior by letting the variance tend towards an *a priori* determined value $\eta$ if $\boldsymbol{x}_0$ tends away from the training data. Let $\{\mathbf{c}_i\}_{i=1}^L$ be points in $\mathbb{R}^D$ that represent the training data, akin to inducing points in sparse GPs [Snelson and Ghahramani, 2006]. Then define $\delta(\boldsymbol{x}_0) = \min_i \|\mathbf{c}_i - \boldsymbol{x}_0\|$ and

$$\hat{\sigma}^2(\boldsymbol{x}_0) = \big(1 - \nu(\delta(\boldsymbol{x}_0))\big)\hat{\sigma}_\theta^2 + \eta\nu(\delta(\boldsymbol{x}_0)), \tag{5}$$

where $\nu : [0, \infty) \mapsto [0, 1]$ is a surjectively increasing function. Then the variance estimate will go to $\eta$ as $\delta \to \infty$ at a rate determined by $\nu$. In practice, we choose $\nu$ to be a scaled-and-translated sigmoid function: $\nu(x) = \text{sigmoid}((x + a)/\gamma)$, where $\gamma$ is a free parameter we optimize during training and $a \approx -6.9077\gamma$ to ensure that $\nu(0) \approx 0$. The inducing points $\mathbf{c}_i$ are initialized with $k$-means and optimized during training. This choice of architecture is similar to that attained by posterior Gaussian processes when the associated covariance function is stationary. It is indeed the behavior of these established models that we aim to mimic with Eq. 5.

# 4 Experiments

## 4.1 Regression

To test our methodologies we conduct multiple experiments in various settings. We compare our method to state-of-the-art methods for quantifying uncertainty: Bayesian neural network (BNN) [Hernández-Lobato and Adams, 2015], Monte Carlo Dropout (MC-Dropout) [Gal and Ghahramani, 2016] and Deep Ensembles (Ens-NN) [Lakshminarayanan et al., 2017]. Additionally we compare to two baseline methods: standard mean-variance neural network (NN) [Nix and Weigend, 1994] and GPs (sparse GPs (SGP) when standard GPs are not applicable) [Rasmussen and Williams, 2006]. We refer to our own method(s) as *Combined*, since we apply all the methodologies described in Sec. 3. Implementation details and code can be found in the supplementary material. Strict comparisons of the models should be carefully considered; having two seperate networks to model mean and variance seperately (as NN, Ens-NN and Combined) means that all the predictive uncertainty, *i.e.* both aleatoric and episteminc, is modeled by the variance networks alone. BNN and MC-Dropout have a higher emphasis on modeling epistemic uncertainty, while GPs have the cleanest separation of noise and model uncertainty estimation. Despite the methods quantifying different types of uncertainty, their results can still be ranked by test set log-likelihood, which is a proper scoring function.

**Toy regression.** We first return to the toy problem of Sec. 1, where we consider 500 points from $y = \boldsymbol{x} \cdot \sin(\boldsymbol{x}) + 0.3 \cdot \epsilon_1 + 0.3 \cdot \boldsymbol{x} \cdot \epsilon_2$, with $\epsilon_1, \epsilon_2 \sim \mathcal{N}(0, 1)$. In this example, the variance is *heteroscedastic*, and models should estimate larger variance for larger values of $\boldsymbol{x}$. The results[6] can be seen in Figs. 2 and 3. Our approach is the only one to satisfy all of the following: capture the heteroscedasticity, extrapolate high variance outside data region and not underestimating within.

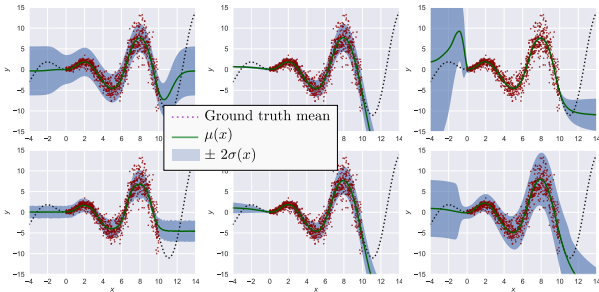 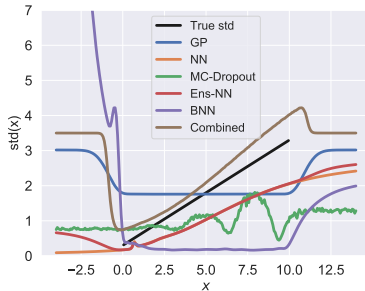

Figure 2: From top left to bottom right: GP, NN, BNN, MC-Dropout, Ens-NN, Combined.

Figure 3: Standard deviation estimates as a function of $\boldsymbol{x}$.

**Variance calibration.** To our knowledge, no benchmark for quantifying variance estimation exists. We propose a simple dataset with known uncertainty information. More precisely, we consider weather data from over 130 years.[7] Each day the maximum temperature is measured, and the uncertainty is then given as the variance in temperature over the 130 years. The fitted models can be seen in Fig. 4. Here we measure performance by calculating the mean error in uncertainty: $\text{Err} = \frac{1}{N} \sum_{i=1}^{N} |\sigma_{\text{true}}^2(\boldsymbol{x}_i) - \sigma_{\text{est}}^2(\boldsymbol{x}_i)|$. The numbers are reported above each fit. We observe that our Combined model achieves the lowest error of all the models, closely followed by Ens-NN and GP. Both NN, BNN and MC-Dropout all severely underestimate the uncertainty.

**Ablation study.** To determine the influence of each methodology from Sec. 3, we experimented with four UCI regression datasets (Fig. 5). We split our contributions in four: the locality sampler (LS), the mean-variance split (MV), the inverse-gamma prior (IG) and the extrapolating architecture (EX). The combined model includes all four tricks. The results clearly shows that LS and IG methodologies has the most impact on test set log likelihood, but none of the methodologies perform worse than the baseline model. Combined they further improves the results, indicating that the proposed methodologies are complementary.

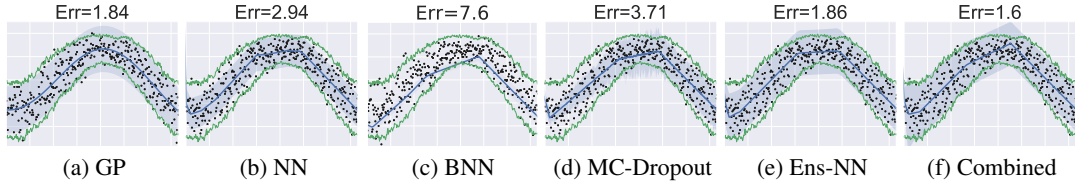

|  |  |  |  |  |  |
|---|---|---|---|---|---|
| Err=1.84 | Err=2.94 | Err=7.6 | Err=3.71 | Err=1.86 | Err=1.6 |
| (a) GP | (b) NN | (c) BNN | (d) MC-Dropout | (e) Ens-NN | (f) Combined |

Figure 4: Weather data with uncertainties. Dots are datapoints, green lines are the true uncertainty, blue curves are mean predictions and the blue shaded areas are the estimated uncertainties.

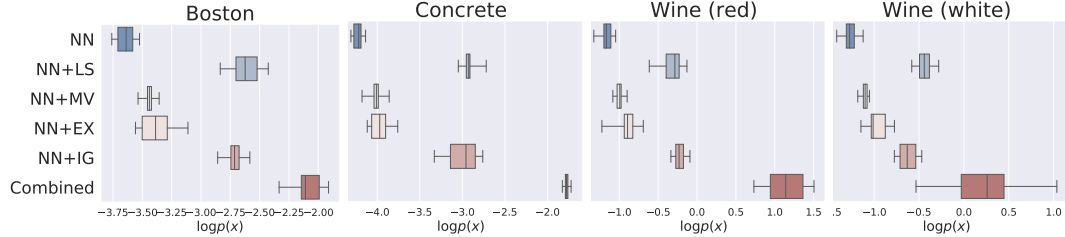

Figure 5: The complementary methodologies from Sec. 3 evaluated on UCI benchmark datasets.

**UCI benchmark.** We now follow the experimental setup from Hernández-Lobato and Adams [2015], by evaluating models on a number of regression datasets from the UCI machine learning database. Additional to the standard benchmark, we have added 4 datasets. Test set log-likelihood can be seen in Table 1, and the corresponding RMSE scores can be found in the supplementary material.

Our *Combined* model performs best on 10 of the 13 datasets. For the small *Boston* and *Yacht* datasets, the standard GP performs the best, which is in line with the experience that GPs perform well when data is scarce. On these datasets our model is the best-performing neural network. On the *Energy* and *Protein* datasets Ens-NN perform the best, closely followed by our Combined model. One clear advantage of our model compared to Ens-NN is that we only need to train one model, whereas Ens-NN need to train 5+ (see the supplementary material for training times for each model). The worst performing model in all cases is the baseline NN model, which clearly indicates that the usual tools for *mean* estimation does not carry over to *variance* estimation.

**Active learning.** The performance of active learning depends on predictive uncertainty [Settles, 2009], so we use this to demonstrate the improvements induced by our method. We use the same network architectures and datasets as in the UCI benchmark. Each dataset is split into: 20% train, 60% pool and 20% test. For each active learning iteration, we first train a model, evaluate the performance on the test set and then estimate uncertainty for all datapoints in the pool. We then select the $n$ points with highest variance (corresponding to highest entropy [Houlsby et al., 2012]) and add these to the

|  | $N$ | $D$ | GP | SGP | NN | BNN | MC-Dropout | Ens-NN | Combined |
|---|---|---|---|---|---|---|---|---|---|
| Boston | 506 | 13 | $\mathbf{-1.76 \pm 0.3}$ | $-1.85 \pm 0.25$ | $-3.64 \pm 0.09$ | $-2.59 \pm 0.11$ | $-2.51 \pm 0.31$ | $-2.45 \pm 0.25$ | $-2.09 \pm 0.09$ |
| Carbon | 10721 | 7 | - | $3.74 \pm 0.53$ | $-2.03 \pm 0.14$ | $-1.1 \pm 1.76$ | $-1.08 \pm 0.05$ | $-0.44 \pm 7.28$ | $\mathbf{4.35 \pm 0.16}$ |
| Concrete | 1030 | 8 | $-2.13 \pm 0.14$ | $-2.29 \pm 0.12$ | $-4.23 \pm 0.07$ | $-3.31 \pm 0.05$ | $-3.11 \pm 0.12$ | $-3.06 \pm 0.32$ | $\mathbf{-1.78 \pm 0.04}$ |
| Energy | 768 | 8 | $-1.85 \pm 0.34$ | $-2.22 \pm 0.15$ | $-3.78 \pm 0.04$ | $-2.07 \pm 0.08$ | $-2.01 \pm 0.11$ | $\mathbf{-1.48 \pm 0.31}$ | $-1.68 \pm 0.13$ |
| Kin8nm | 8192 | 8 | - | $2.01 \pm 0.02$ | $-0.08 \pm 0.02$ | $0.95 \pm 0.08$ | $0.95 \pm 0.15$ | $1.18 \pm 0.03$ | $\mathbf{2.49 \pm 0.07}$ |
| Naval | 11934 | 16 | - | - | $3.47 \pm 0.21$ | $3.71 \pm 0.05$ | $3.80 \pm 0.09$ | $5.55 \pm 0.05$ | $\mathbf{7.27 \pm 0.13}$ |
| Power plant | 9568 | 4 | - | $-1.9 \pm 0.03$ | $-4.26 \pm 0.14$ | $-2.89 \pm 0.01$ | $-2.89 \pm 0.14$ | $-2.77 \pm 0.04$ | $\mathbf{-1.19 \pm 0.03}$ |
| Protein | 45730 | 9 | - | - | $-2.95 \pm 0.09$ | $-2.91 \pm 0.00$ | $-2.93 \pm 0.14$ | $\mathbf{-2.80 \pm 0.02}$ | $-2.83 \pm 0.05$ |
| Superconduct | 21263 | 81 | - | $-4.07 \pm 0.01$ | $-4.92 \pm 0.10$ | $-3.06 \pm 0.14$ | $-2.91 \pm 0.19$ | $-3.01 \pm 0.05$ | $\mathbf{-2.43 \pm 0.05}$ |
| Wine (red) | 1599 | 11 | $0.96 \pm 0.18$ | $-0.08 \pm 0.01$ | $-1.19 \pm 0.11$ | $-0.98 \pm 0.01$ | $-0.94 \pm 0.01$ | $-0.93 \pm 0.09$ | $\mathbf{1.21 \pm 0.23}$ |
| Wine (white) | 4898 | 11 | - | $-0.14 \pm 0.05$ | $-1.29 \pm 0.09$ | $-1.41 \pm 0.17$ | $-1.26 \pm 0.01$ | $-0.99 \pm 0.06$ | $\mathbf{0.40 \pm 0.42}$ |
| Yacht | 308 | 7 | $\mathbf{0.16 \pm 1.22}$ | $-0.38 \pm 0.32$ | $-4.12 \pm 0.17$ | $-1.65 \pm 0.05$ | $-1.55 \pm 0.12$ | $-1.18 \pm 0.21$ | $-0.07 \pm 0.05$ |
| Year | 515345 | 90 | - | - | $-5.21 \pm 0.87$ | $-3.97 \pm 0.34$ | $-3.78 \pm 0.01$ | $-3.42 \pm 0.02$ | $\mathbf{-3.01 \pm 0.14}$ |

Table 1: Dataset characteristics and tests set log-likelihoods for the different methods. A - indicates the model was infeasible to train. Bold highlights the best results.

training set. We set $n = 1\%$ of the initial pool size. This is repeated 10 times, such that the last model is trained on 30%. We repeat this on 10 random training-test splits to compute standard errors.

Fig. 6,show the evolution of average RMSE for each method during the data collection process for the *Boston*, *Superconduct* and *Wine (white)* datasets (all remaining UCI datasets are visualized in the supplementary material). In general, we observe two trends. For some datasets we observe that our *Combined* model outperforms all other models, achieving significantly faster learning. This indicates that our model is better at predicting the uncertainty of the data in the pool set. On datasets where the sampling process does not increase performance, we are on par with other models.

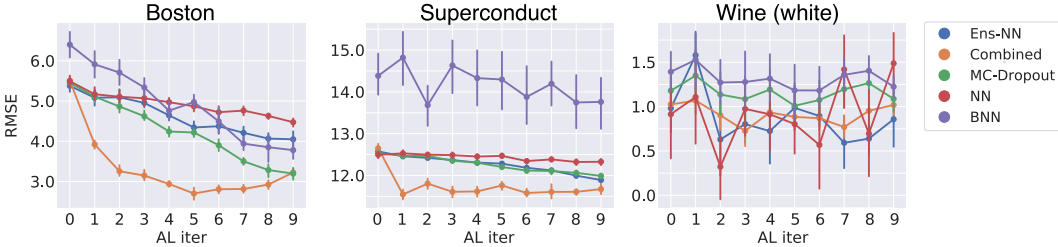

Figure 6: Average test set RMSE and standard errors in active learning. The remaining datasets are shown in the supplementary material.

## 4.2 Generative models

To show a broader application of our approach, we also explore it in the context of generative modeling. We focus on variational autoencoders (VAEs) [Kingma and Welling, 2013, Rezende et al., 2014] that are popular deep generative models. A VAE model the generative process:

$$p(\boldsymbol{x}) = \int p_\theta(\boldsymbol{x}|\boldsymbol{z})p(\boldsymbol{z})\mathrm{d}\boldsymbol{z}, \quad p_\theta(\boldsymbol{x}|\boldsymbol{z}) = \mathcal{N}\big(\boldsymbol{x}|\mu_\theta(\boldsymbol{z}), \sigma_\theta^2(\boldsymbol{z})\big) \quad \text{or} \quad p_\theta(\boldsymbol{x}|\boldsymbol{z}) = \mathcal{B}\big(\boldsymbol{x}|\mu_\theta(\boldsymbol{z})\big), \quad (6)$$

where $p(\boldsymbol{z}) = \mathcal{N}(\boldsymbol{0}, \mathbb{I}_d)$. This is trained by introducing a variational approximation $q_\phi(\boldsymbol{z}|\boldsymbol{x}) = \mathcal{N}(\boldsymbol{z}|\mu_\phi(\boldsymbol{x}), \sigma_\phi^2(\boldsymbol{x}))$ and then jointly training $p_\theta$ and $q_\phi$. For our purposes, it suffcient to note that a VAE estimates both a mean and a variance function. Thus using standard training methods, the same problems arise as in the regression setting. Mattei and Frellsen [2018] have recently shown that estimating a VAE is ill-posed unless the variance is bounded from below. In the literature, we often find that

**1.** Variance networks are avoided by using a Bernoulli distribution, even if data is not binary.

**2.** Optimizing VAEs with a Gaussian posterior is considerably harder than the Bernoulli case. To overcome this, the variance is often set to a constant *e.g.* $\sigma^2(\boldsymbol{z}) = 1$. The consequence is that the log-likelihood reconstruction term in the ELBO collapses into an L2 reconstruction term.

**3.** Even though the generative process is given by Eq. 6, samples shown in the literature are often reduced to $\tilde{\boldsymbol{x}} = \mu(\boldsymbol{z}), \boldsymbol{z} \sim \mathcal{N}(\boldsymbol{0}, \mathbb{I})$. This is probably due to the wrong/meaningless variance term.

We aim to fix this by training the posterior variance $\sigma_\theta^2(\boldsymbol{z})$ with our Combined method. We do not change the encoder variance $\sigma_\phi^2(\boldsymbol{x})$ and leave this to future study.

**Artificial data.** We first evaluate the benefits of more reliable variance networks in VAEs on artificial data. We generate data inspired by the two moon dataset[8], which we map into four dimensions. The mapping is thoroughly described in the supplementary material, and we emphasize that we have deliberately used mappings that MLP's struggle to learn, thus with a low capacity network the only way to compensate is to learn a meaningful variance function.

In Fig. 7 we plot pairs of output dimensions using 5000 generated samples. For all pairwise combinations we refer to the supplementary material. We observe that samples from our Comb-VAE capture the data distribution in more detail than a standard VAE. For VAE the variance seems to be

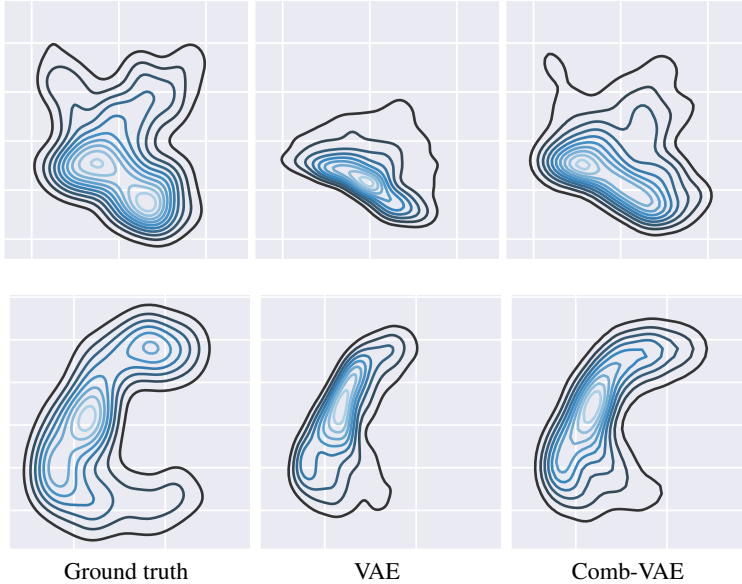

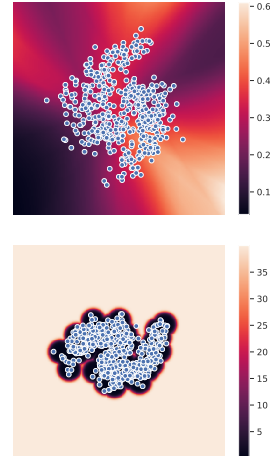

Figure 8: Variance estimates in latent space for standard VAE (top) and our Comb-VAE (bottom). Blue points are the encoded training data.

Figure 7: The ground truth and generated distributions. *Top*: $x_1$ vs. $x_2$. *Bottom*: $x_2$ vs $x_3$.

|  |  | MNIST | FashionMNIST | CIFAR10 | SVHN |
|---|---|---|---|---|---|
| ELBO | VAE | $2053.01 \pm 1.60$ | $1506.31 \pm 2.71$ | $1980.84 \pm 3.32$ | $3696.35 \pm 2.94$ |
|  | Comb-VAE | $\mathbf{2152.31 \pm 3.32}$ | $\mathbf{1621.29 \pm 7.23}$ | $\mathbf{2057.32 \pm 8.13}$ | $3701.41 \pm 5.84$ |
| $\log p(x)$ | VAE | $1914.77 \pm 2.15$ | $1481.38 \pm 3.68$ | $1809.43 \pm 10.32$ | $3606.28 \pm 2.75$ |
|  | Comb-VAE | $\mathbf{2018.37 \pm 4.35}$ | $\mathbf{1567.23 \pm 4.82}$ | $\mathbf{1891.39 \pm 20.21}$ | $3614.39 \pm 7.91$ |

Table 2: Generative modeling of 4 datasets. For each dataset we report training ELBO and test set log-likelihood. The standard errors are calculated over 3 trained models with random initialization.

underestimated, which is similar to the results from regression. The poor sample quality of a standard VAE can partially be explained by the arbitrariness of decoder variance function $\sigma^2(\boldsymbol{z})$ away from data. In Fig. 8, we calculated the accumulated variance $\sum_{j=1}^{D} \sigma_j^2(\boldsymbol{z})$ over a grid of latent points. We clearly see that for the standard VAE, the variance is low where we have data and arbitrary away from data. However, our method produces low-variance region where the two half moons are and a high variance region away from data. We note that Arvanitidis et al. [2018] also dealt with the problem of arbitrariness of the decoder variance. However their method relies on post-fitting of the variance, whereas ours is fitted during training. Additionally, we note that [Takahashi et al., 2018] also successfully modeled the posterior of a VAE as a Student t-distribution similar to our proposed method, but without the extrapolation and different training procedure.

**Image data.** For our last set of experiments we fitted a standard VAE and our Comb-VAE to four datasets: MNIST, FashionMNIST, CIFAR10, SVHN. We want to measure whether there is an improvement to generative modeling by getting better variance estimation. The details about network architecture and training can be found in the supplementary material. Training set ELBO and test set log-likelihoods can be viewed in Table 2. We observe on all datasets that, on average tighter bounds and higher log-likelihood are achieved, indicating that we better fit the data distribution. We quantitatively observe (see Fig. 9) that variance has a more local structure for Comb-VAE and that the variance reflects the underlying latent structure.

## 5  Discussion & Conclusion

While variance networks are commonly used for modeling the predictive uncertainty in regression and in generative modeling, there have been no systematic studies of how to fit these to data. We have demonstrated that tools developed for fitting *mean* networks to data are subpar when applied to

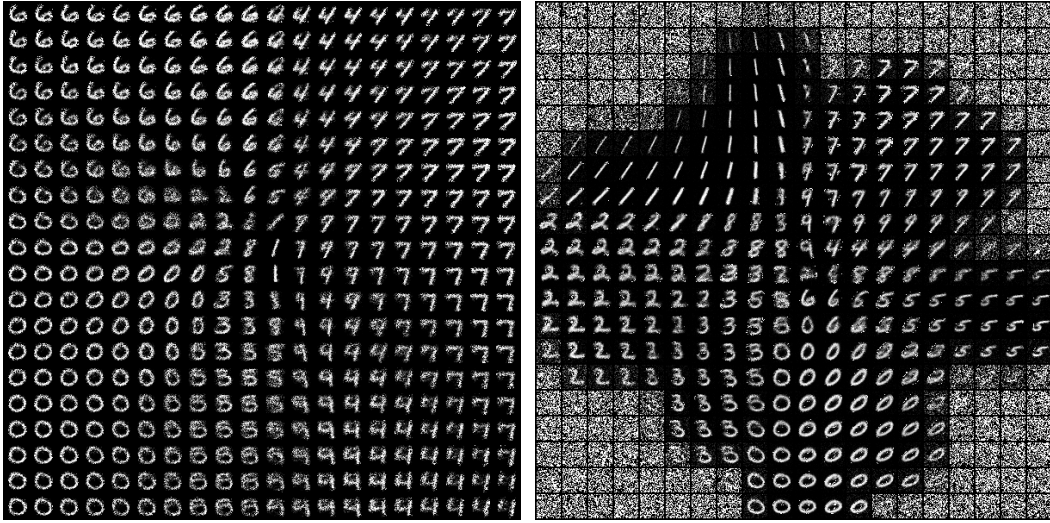

Figure 9: Generated MNIST images on a grid in latent space using the standard variance network (left) and proposed variance network (right).

*variance* estimation. The key underlying issue appears to be that it is not feasible to estimate both a mean and a variance at the same time, when data is scarce.

While it is beneficial to have separate estimates of both *epistemic* and *aleatoric* uncertainty, we have focused on *predictive uncertainty*, which combine the two. This is a lesser but more feasible goal.

We have proposed a new mini-batching scheme that samples locally to ensure that variances are better defined during model training. We have further argued that variance estimation is more meaningful when conditioned on the mean, which implies a change to the usual training procedure of joint mean-variance estimation. To cope with data scarcity we have proposed a more robust likelihood that model a distribution over the variance. Finally, we have highlighted that variance networks need to extrapolate differently from mean networks, which implies architectural differences between such networks. We specifically propose a new architecture for variance networks that ensures similar variance extrapolations to posterior Gaussian processes from stationary priors.

Our methodologies depend on algorithms that computes Euclidean distances. Since these often break down in high dimensions, this indicates that our proposed methods may not be suitable for high dimensional data. Since we mostly rely on nearest neighbor computations, that empirical are known to perform better in high dimensions, our methodologies may still work in this case. Interestingly, the very definition of variance is dependent on Euclidean distance and this may indicate that variance is inherently difficult to estimate for high dimensional data. This could possible be circumvented through a learned metric.

Experimentally, we have demonstrated that proposed methods are complementary and provide significant improvements over state-of-the-art. In particular, on benchmark data we have shown that our method improves upon the test set log-likelihood without improving the RMSE, which demonstrate that the uncertainty is a significant improvement over current methods. Another indicator of improved uncertainty estimation is that our method speeds up active learning tasks compared to state-of-the-art. Due to the similarities between active learning, Bayesian optimization, and reinforcement learning, we expect that our approach carries significant value to these fields as well. Furthermore, we have demonstrated that variational autoencoders can be improved through better generative variance estimation. Finally, we note that our approach is directly applicable alongside ensemble methods, which may further improve results.

**Acknowledgements.** This project has received funding from the European Research Council (ERC) under the European Union's Horizon 2020 research and innovation programme (grant agreement nº 757360). NSD, MJ and SH were supported in part by a research grant (15334) from VILLUM FONDEN. We gratefully acknowledge the support of NVIDIA Corporation with the donation of GPU hardware used for this research.

## Footnotes

[3]By convention, we say that the nearest neighbor of a point is the point itself.

[4]Degrees of freedom here refers to the parameters in a Gamma distribution – the distribution of variance estimators under Gaussian likelihood. Degrees of freedom in general is a quite elusive quantity in regression problems.

[5]This means $y \sim F$, where $F = \mu + \sigma t(\nu)$. The explicit density can be found in the supplementary material.

[6]The standard deviation plotted for *Combined*, is the root mean of the inverse-Gamma.

[7]`https://mrcc.illinois.edu/CLIMATE/Station/Daily/StnDyBTD2.jsp`

[8]https://scikit-learn.org/stable/modules/generated/sklearn.datasets.make_moons.

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
