[Supplementary Material]

# Reliable training and estimation of variance networks -Supplementary Material-

**Nicki S. Detlefsen**[*][†]
nsde@dtu.dk

**Martin Jørgensen**[*][†]
marjor@dtu.dk

**Søren Hauberg**[†]
sohau@dtu.dk

## Abstract

This paper contains the supplementary material to the paper *Reliable training and estimation of variance networks*. The current document contains the following sections: 1. Further experimental results, 2. Additional details about the locality sampler, 3. Details about the network architecture and training for the generative experiments, 4. explanation how the generative toy data was generated.

## 1 Further results

**Gradient experiments** In Fig. 1 we have plotted the sparsity index and variance of gradient for both the mean (top row) and variance function (bottom row). We do this for both normal mini-batching and our proposed locality sampler. Sparsity is measured as $\ell^0_{0.001}(\nabla) = \{j, \nabla_j \leq 0.001\}$ and the sparsity index is then given by $\text{SI} = \frac{\ell^0_{0.001}(\nabla)}{|\nabla|}$. We observe for the mean function, that the sparsity index and variance is similar for the two methods, indicating that our locality sampler does not improve on the fitting of the mean function, as expected. However for the variance function we see a clear gap in sparsity and variance, indicating that our locality sampler gives more local and stable updates to variance networks.

**UCI benchmark (RMSE)** In Table 1 the test set RMSE results for the UCI regression benchmark can be seen. We clearly observe that all neural network based methods achieve nearly identical RMSE for all datasets, indicating that the mean function is similarly trained for all the methods.

| | $N$ | $D$ | GP | SGP | NN | BNN | MC-Dropout | Ens-NN | Combined |
|---|---|---|---|---|---|---|---|---|---|
| Boston | 506 | 13 | $2.79 \pm 0.52$ | $2.98 \pm 0.55$ | $4.45 \pm 1.41$ | $3.45 \pm 0.87$ | $3.01 \pm 0.99$ | $3.33 \pm 1.33$ | $3.11 \pm 0.35$ |
| Carbon | 10721 | 7 | - | $1.01 \pm 0.01$ | $0.41 \pm 0.0$ | $0.18 \pm 0.1$ | $0.29 \pm 0.0$ | $0.41 \pm 0.0$ | $0.35 \pm 0.01$ |
| Concrete | 1030 | 8 | $6.03 \pm 0.59$ | $6.45 \pm 0.64$ | $7.71 \pm 1.32$ | $5.78 \pm 0.21$ | $5.33 \pm 0.65$ | $5.65 \pm 0.55$ | $5.75 \pm 0.41$ |
| Energy | 768 | 8 | $1.98 \pm 0.76$ | $2.12 \pm 0.56$ | $1.67 \pm 0.44$ | $1.89 \pm 0.04$ | $1.69 \pm 0.19$ | $2.13 \pm 0.46$ | $1.70 \pm 0.21$ |
| Kin8nm | 8192 | 8 | - | $0.08 \pm 0.0$ | $0.21 \pm 0.01$ | $0.18 \pm 0.02$ | $0.12 \pm 0.01$ | $0.01 \pm 0.01$ | $0.12 \pm 0.01$ |
| Naval | 11934 | 16 | - | - | $0.01 \pm 0.0$ | $0.01 \pm 0.0$ | $0.01 \pm 0.0$ | $0.01 \pm 0.0$ | $0.01 \pm 0.0$ |
| Power plant | 9568 | 4 | - | $4.65 \pm 0.12$ | $4.23 \pm 0.33$ | $4.12 \pm 0.45$ | $4.13 \pm 0.13$ | $4.11 \pm 0.21$ | $4.12 \pm 0.13$ |
| Protein | 45730 | 9 | - | - | $4.38 \pm 0.07$ | $4.67 \pm 0.94$ | $4.19 \pm 0.08$ | $4.36 \pm 0.07$ | $4.52 \pm 0.19$ |
| Superconduct | 21263 | 81 | - | $11.32 \pm 0.38$ | $11.73 \pm 0.46$ | $11.07 \pm 1.7$ | $11.44 \pm 0.39$ | $11.63 \pm 0.49$ | $11.65 \pm 0.65$ |
| Wine (red) | 1599 | 11 | $0.88 \pm 0.06$ | $0.65 \pm 0.04$ | $0.66 \pm 0.06$ | $0.69 \pm 0.41$ | $0.64 \pm 0.06$ | $0.67 \pm 0.06$ | $0.68 \pm 0.11$ |
| Wine (white) | 4898 | 11 | - | $0.65 \pm 0.03$ | $0.67 \pm 0.04$ | $0.68 \pm 0.32$ | $0.71 \pm 0.04$ | $0.78 \pm 0.04$ | $0.72 \pm 0.09$ |
| Yacht | 308 | 7 | $0.42 \pm 0.21$ | $0.72 \pm 0.21$ | $1.63 \pm 0.61$ | $1.05 \pm 0.11$ | $1.11 \pm 0.48$ | $1.58 \pm 0.58$ | $1.27 \pm 0.11$ |
| Year | 515345 | 90 | - | - | $12.47 \pm 0.96$ | $9.01 \pm 0.45$ | $8.92 \pm 0.23$ | $8.88 \pm 0.13$ | $8.85 \pm 0.05$ |

Table 1: Dataset characteristics and RMSE for the different methods. A - indicates the models was infeasible to train.

**Timings of models** In Table 2 we show the average computation time for each model. The experiments was conducted with an Intel Xeon E5-2620v4 CPU and Nvidia GTX TITAN X GPU.

---

[*]Equal contribution

[†]Section for Cognitive Systems, Technical University of Denmark

Figure 1: *Left*: Sparsity index for mean (top) and variance (bottom) network. *Right*: Variance of gradient for mean (top) and variance (bottom) network. The variance network was disabled for the first 2500 iterations, to warm up the mean function to secure convergence.

We note that our model suffers from long computations for very large datasets, mainly due to the computation of the neighborhood graph in the locality sampler. This could be reduced by using fast approximative method for k-nearest-neighbor and by reducing data dimensionality *e.g.* PCA dimensionality reduction.

| | N | D | GP | SGP | NN | BNN | MC-Dropout | Ens-NN | Combined |
|---|---|---|---|---|---|---|---|---|---|
| Boston | 506 | 13 | 8.37 +- 2.92 | 91.86 +- 30.12 | 94.04 +- 2.24 | 81.32 +- 1.83 | 98.08 +- 2.0 | 479.1 +- 12.49 | 93.39 +- 1.82 |
| Carbon | 10721 | 7 | - | 192.57 +- 72.28 | 90.05 +- 3.4 | 80.95 +- 2.04 | 98.62 +- 1.85 | 439.45 +- 23.01 | 123.61 +- 2.84 |
| Concrete | 1030 | 8 | 7.48 +- 1.2 | 173.73 +- 4.0 | 92.91 +- 4.17 | 81.02 +- 2.04 | 97.94 +- 1.84 | 468.94 +- 10.81 | 97.65 +- 6.4 |
| Energy | 768 | 8 | 10.48 +- 4.12 | 121.39 +- 52.2 | 92.91 +- 2.14 | 80.92 +- 2.14 | 97.86 +- 2.13 | 475.04 +- 12.61 | 93.01 +- 1.29 |
| Kin8nm | 8192 | 8 | - | 1526.29 +- 20.28 | 92.22 +- 4.65 | 80.65 +- 2.11 | 97.85 +- 2.87 | 459.81 +- 27.66 | 123.15 +- 2.86 |
| Navel | 11934 | 16 | - | 9.79 +- 0.23 | 91.2 +- 3.28 | 82.97 +- 1.84 | 98.64 +- 1.89 | 482.74 +- 8.54 | 136.25 +- 3.29 |
| Power | 9568 | 4 | - | 1267.82 +- 783.28 | 92.23 +- 3.28 | 81.29 +- 1.98 | 98.26 +- 1.87 | 472.39 +- 9.73 | 118.26 +- 1.86 |
| Protein | 45730 | 9 | - | - | 138.49 +- 2.51 | 124.72 +- 1.86 | 140.73 +- 3.32 | 707.69 +- 11.02 | 658.63 +- 11.75 |
| Superconduct | 21263 | 81 | - | 313.9 +- 2.9 | 95.27 +- 3.43 | 90.71 +- 1.86 | 90.25 +- 1.67 | 477.57 +- 11.91 | 235.72 +- 3.8 |
| Wine (red) | 1599 | 11 | 35.33 +- 19.09 | 262.69 +- 10.14 | 93.51 +- 2.09 | 80.67 +- 2.04 | 97.78 +- 2.58 | 416.02 +- 23.9 | 130.61 +- 7.91 |
| Wine (white) | 4898 | 11 | - | 781.13 +- 8.2 | 91.97 +- 2.41 | 80.94 +- 1.75 | 97.61 +- 2.73 | 451.65 +- 19.04 | 99.44 +- 1.33 |
| Yacht | 308 | 7 | 0.93 +- 0.32 | 22.74 +- 11.98 | 92.82 +- 3.47 | 81.0 +- 1.92 | 98.27 +- 1.68 | 422.9 +- 35.85 | 105.33 +- 2.32 |
| Year | 515345 | 90 | - | - | 139.45 +- 6.15 | 643.96 +- 4.17 | 77.96 +- 2.09 | 725.27 +- 20.08 | 2453.62 +- 18.7 |

Table 2: Timings(s) for the different models evaluated on the UCI benchmark

**Ablation study (RMSE)**    In Fig. 2 we have plotted the RMSE for different combinations of our methodologies. Since the RMSE is only influenced by how well $\mu(x)$ is fitted, the difference in log likelihood that we observe between the models (see paper) must be explained by how well $\sigma^2(x)$ is fitted.

**Active learning**    In Figs. 3 and 4 we respective shows the progress of RMSE and log likelihood on all 13 dataset. We observe that for some of the datasets (Boston, Superconduct, Power) our proposed Combine model achieves faster learning than other methods. On all other datasets we are equally good as the best performing model.

Figure 2: Our contributions evaluated on four different UCI benchmark datasets.

Figure 3: Average test RMSE and standard errors in the active learning experiments for all datasets.

Figure 4: Average test log likelihood and standard errors in the active learning experiments for all datasets.

**Generative modeling toy data** In Fig. 6 we show marginal distribution, pairwise pointplots and pairwise joint distribution for our artificially dataset used in the generative setting. Top row show the ground true data, middle row shows reconstructions and samples from standard VAE model and bottom row show reconstructions and samples from our proposed Comb-VAE model. We observe that reconstruction are similar for the two models, but the quality of the generative samples are much better for Comb-VAE.

**Generative modeling of image data** In Fig. 5 we show a meshgrid of samples from VAE and Comb-VAE on the MNIST dataset. We clearly see how proper extrapolation of variance in Comb-VAE can be used to "mask" when we are inside our data region and when we are outside. For standard VAE we observe a near constant variance added to the images.

(a) VAE                                                 (b) Comb-VAE

Figure 5: Meshgrid of latent space. For each subplot we sampled a mesh grid $[-4, 4] \times [-4, 4]$ of $[50, 50]$ points, which we used to generate samples from.

## 2 On the parametrization of the $t$-distributed predictive distribution

We parametrize the Student-$t$ distribution by letting the variance $\sigma^2$ have an inverse-Gamma distribution with shape and scale parameters $\alpha$ and $\beta$. We use that if $\sigma^2 \sim \text{INV-GAMMA}(\alpha, \beta)$ then $\frac{1}{\sigma^2} \sim \Gamma(\alpha, \beta)$. Then

$$
\begin{aligned}
p(y|\mu, \alpha, \beta) &= \int_0^\infty \mathcal{N}(y|\mu, \sigma^2) \frac{\beta^\alpha}{\Gamma(\alpha)} (\sigma^2)^{-(\alpha+1)} \exp(-\frac{\beta}{\sigma^2}) \mathrm{d}\sigma^2 \\
&= \frac{\beta^\alpha}{\Gamma(\alpha)\sqrt{2\pi}} \int_0^\infty (\sigma^2)^{-(\alpha+1)-\frac{1}{2}} \exp(-\frac{1}{\sigma^2}(\beta + \frac{1}{2}(y-\mu)^2)) \mathrm{d}\sigma^2 \\
&= \frac{\beta^\alpha}{\Gamma(\alpha)\sqrt{2\pi}} \frac{\Gamma(\alpha + \frac{1}{2})}{\left(\beta + \frac{1}{2}(y-\mu)^2\right)^{\alpha+\frac{1}{2}}},
\end{aligned}
$$

where we substituted the variable $\sigma^2$ with $\frac{1}{\sigma^2}$ and used that the remaining was a Gamma integral.

## 3 Parameters of the locality sampler

In Algorithm 1 a pseudoimplementation of our proposed locality sampler can be seen. The two important parameters in this algorithm are the primary sampling units (psu) and secondary sampling units (ssu). In Fig. 7 and 8 we visually show the effect of these two parameters.

For all our experiments we set psu=3 and ssu=40 when we are training the mean function and for variance function we set psu=1 and ssu=10.

(a) Ground true

(b) VAE (R)

(c) VAE (S)

(d) Comb-VAE (R)

(e) Comb-VAE (S)

Figure 6: Pairwise plots between all sets of variables for our artificially dataset in the generative setting.

---

**Algorithm 1** Locality-sampler

---

**Input** $N$ datapoints, a metric $d$ on feature space $\mathbb{R}^{\mathbb{D}}$, integers $m, n, k$.

1: For each datapoint calculate the $k$ nearest neighbors under the metric $d$.
2: Sample $m$ primary sampling units with uniform probability without replacement among all $N$ units.
3: For each of the primary sampling units sample $n$ secondary sampling units among the primary sampling units $k$ nearest neighbors with uniform probability without replacement.

**Output** All secondary sampling units is a sample of at most $m \cdot n$ points. If a new sample is needed repeat from Step 2.

---

Figure 7: The effect of changing the size of the primary sampling unit. From top to bottom: $psu = [1, 2, 3]$. Each column corresponds to a sample from the locality sampler.

Figure 8: The effect of changing the size of the secondary sampling unit. From top to bottom: $ssu = [10, 50, 100]$. Each column corresponds to a sample from the locality sampler.

## 4   Implementation details for regression experiments

All neural network based models were implemented in Pytorch [Paszke et al., 2017], except for the Baysian neural network which was implemented in Tensorflow [Abadi et al., 2015]. GP models where implemented in GPy [GPy, since 2012]. Below we have stated details for all models:

**GP** Fitted using a ARD kernel and with default settings of GPy.

**SGP** Fitted using a ARD kernel and with default settings of GPy. Number of inducing points was set to $\min(500, |\mathcal{D}_{train}|)$.

**NN** Model use two networks, one for the predictive mean and one for the predictive variance. Model was trained to optimize the log-likelihood of data.

**BNN** We use a standard factored Gaussian prior for the weights and use the Flipout approximation [Wen et al., 2018] for the variational approximation.

**MC-Dropout** Model use a single network, where we place dropout on all weights. The dropout weight was set to 0.05. The model was trained to optimize the RMSE of data.

**Ens-NN** Model consist of an ensemble of 5 individual NN models, each modeled as two individual networks. Each are trained to optimize the log-likelihood of data. Only difference between ensemble models is initialization.

**Combined** Model use three networks, one for the mean function, one for the $\alpha$ parameter and one for the $\beta$ parameter. We set the number of inducing points to $\min(500, |\mathcal{D}_{train}|)$. For the $\gamma$ in the scaled-and-translated sigmoid function $\nu(x)$ we initialize it to 1.5, and try to minimize it during training. Model was trained to optimize the log-likelihood of data.

For each neural network based approach, we follow the experimental setup [Hernández-Lobato and Adams, 2015]. All individual networks was modeled a single hidden layer MLPs with 50 neurons for all other datasets than "Protein" and "Year" where we use 100 neurons. Except for the output of each network, the activation function used is ReLU. For the output of the mean networks, no activation function is applied. For the output of the variance network, the Softplus activation function is used to secure positive variance. All neural network based models where trained using the Adam optimizer [Kingma and Ba, 2015] with a learning rate of $10^{-1}$ using a batch size of 256. All models were trained for 10.000 iterations.

The code can found here: `https://github.com/SkafteNicki/john`.

## 5 Generative network architecture

Pixel values of the images were scaled to the interval [0,1]. Each pixel is assumed to be i.i.d. Gaussian distributed. For the encoders and decoders we use multilayer perceptron networks, see table below.

|  | Layer 1 | Layer 2 | Layer 3 | Layer 4 |
|---|---|---|---|---|
| $\boldsymbol{\mu}_{encoder}$ | 512 (BN + ReLU) | 256 (BN + ReLU) | 128 (BN + ReLU) | d (BN + Linear) |
| $\boldsymbol{\sigma}^2_{encoder}$ | 512 (BN + ReLU) | 256 (BN + ReLU) | 128 (BN + ReLU) | d (Softplus) |
| $\boldsymbol{\mu}_{decoder}$ | 128 (BN + ReLU) | 256 (BN + ReLU) | 512 (BN + ReLU) | D (ReLU) |
| $\boldsymbol{\sigma}^2_{decoder}$ | 128 (BN + ReLU) | 256 (BN + ReLU) | 512 (BN + ReLU) | D (Softplus) |

The numbers corresponds to the size of the layer and the parenthesis states the used activation function and whether or not batch normalization was used. $D$ indicates the size of the images *i.e.* $D = width \times height \times channels$. For MNIST and FashionMNIST these are 28,28,1 and for CIFAR10 and SVHN these are 32,32,1. $d$ indicates the size of the latent space. For MNIST and FashionMNIST we set $d = 2$ for visualization purpose and for CIFAR10 and SVHN we set $d = 10$ to be able to capture the higher complexity of these datasets.

To train the networks we used the Adam optimizer [Kingma and Ba, 2015] with learning rate $10^{-3}$ and a batch size of 512. We train for 20000 iterations without early stopping. Additionally, we use warm-up for the KL term [Sønderby et al., 2016], by scaling it with $w = \min\left(1, \frac{it}{warmup}\right)$ where $it$ is the current iteration number and $warmup$ was set to half the number of iterations. This secures that we converge to stable reconstructions before introducing too much structure in the latent space.

## 6 Artificial Data

The data considered in Section 4 are generated in the following way: first we sample points in $\mathbb{R}^2$ in a two-moon type way. See details in Algorithm 2. We generate 500 points in this way to establish a

'known' latent space. We then map these to four dimensions $(v_1, v_2, v_3, v_4)$ by

$$v_1(z_1, z_2) = z_1 - z_2 + \epsilon \cdot \sqrt{0.03 + 0.05 \cdot (3 + z_1)}, \tag{1}$$

$$v_2(z_1, z_2) = z_1^2 + \frac{1}{2}z_2 + \epsilon \cdot \sqrt{0.03 + 0.03 \cdot \|\mathbf{z}\|_2}, \tag{2}$$

$$v_3(z_1, z_2) = z_1 z_2 - z_1 + \epsilon \cdot \sqrt{0.03 + 0.05 \cdot \|\mathbf{z}\|_2}, \tag{3}$$

$$v_4(z_1, z_2) = z_1 + z_2 + \epsilon \cdot \sqrt{0.03 + \frac{0.03}{0.2 + \|\mathbf{z}\|_2}}, \tag{4}$$

where all $\epsilon \sim \mathcal{N}(0, 1)$ and independent. A typical dataset from this procedure is shown in Figure 9.

---

**Algorithm 2** Two moon sampler

---

1: Sample $U \sim \text{Bernoulli}(0.5)$.
2: **if** $U = 1$ **then**
3:     Set $c = (0.5, 0)$ and sample $\alpha_1 \sim \text{unif}[0, \pi]$.
4:     Let $z = c + (\cos(\alpha_1), \sin(\alpha_1))$, and sample $\alpha_2 \sim \text{unif}[0, 2\pi]$ and $u \sim \text{unif}[0, 1]$.
5:     Let $z = z + \frac{u}{4} \cdot (\cos(\alpha_2), \sin(\alpha_2))$.
6: **else**
7:     Set $c = (-0.5, 0)$ and sample $\alpha_1 \sim \text{unif}[\pi, 2\pi]$.
8:     Let $z = c + (\cos(\alpha_1), \sin(\alpha_1))$, and sample $\alpha_2 \sim \text{unif}[0, 2\pi]$ and $u \sim \text{unif}[0, 1]$.
9:     Let $z = z + \frac{u}{4} \cdot (\cos(\alpha_2), \sin(\alpha_2))$.
    **return** $z$

---

Figure 9: An example of the two moon data, and its transformation into $\mathbb{R}^4$. Shown as pairwise scatterplots.