[Reviews · NeurIPS 2019]

Reviewer 1



POST-REBUTTAL UPDATE: I have read the rebuttal and other reviews; my score remains unchanged. ======================================================= Originality: The novelty of each proposed trick is limited. However, the combination is novel, well-motivated and supported by the experiments. Quality: The paper is technically sound. Clarity: The paper is very well-written and is easy to follow. The authors clearly pose the problems, provide well-motivated solutions for each of them, and then provide extensive experiments including an ablation study. Significance: The authors test the proposed tricks on a broad set of tasks and outline several other fields that could also benefit. I expect these findings to have a large impact in practical applications. My main concern is regarding the significance of the posed three problems: underestimation of the variance, trivialization of the variance (independence from x) and poor estimation for out-of-domain data (variance does not increase outside the data support). While all of these problems intuitively seem to be common, it would be nice to see a more thorough investigation. Right now these problems are illustrated using the toy data. The improvement in the ablation study can serve as an indirect indication of these problems, however, a direct study of these problems on several different tasks could provide better insight.

Reviewer 2



This is overall a well written paper that highlights weaknesses in a widely used basic building block for building probabilistic models with neural networks: estimating the variance of some (Gaussian) observable. The paper empirically shows that current approaches clearly underestimate variance even in very simple, low-dimensional cases. It also shows that this is true for approaches that otherwise have been shown to improve the “bayesianess” of neural network predictions. I think it is a strength of the paper that the authors concentrate on simple, low-dimensional problems to analyze the problem. Even though this issue has been occasionally mentioned before, I think this highly original work that focuses on this otherwise not sufficiently discussed problem. The authors suggest three distinct approaches to tackle the problem and demonstrate that each of them provide improvements over the current state of affairs, at least in the investigated low- to medium dimensional cases. Each of these approaches is well motivated and described in sufficient detail (especially when considering the supplement). Unfortunately I find the most effective method of these three proposed improvements a bit unsatisfactory because, as the authors point out, it requires significant additional compute resources and introduces complexity into the training. It is also a bit doubtful whether this approach works in cases with high-dimensional input data because it requires finding ‘nearest neighbours’. This aspect was unfortunately not investigated in the submission. In general I think this work can be influential and encourage further research into this particular aspect of neural network modeling and training.

Reviewer 3



Post-Rebuttal Feedback Thank the reviewers for your feedback. I think this is a good paper to appear in NeurIPS. ####################### Uncertainty estimation has always been an important problem. This paper tackles the uncertainty prediction via directly predicting the marginal mean and variances. For assuring the reliability of its uncertainty estimation, the paper presents a series of interesting techniques for training the prediction network, including location-aware mini-batching, mean-variance split training and variance networks. With all these techniques adopted, the paper demonstrates convincing empirical results on its uncertainty estimation. Weakness, I am surprised by the amazing empirical performance and the simplicity of the method. However, many proposed techniques in the paper is not well justified. 1, Overfitting is still a potential issue that might occur. Because the network is simply trained via MLE, it is possible that the network fits all training points perfectly, and predicts zero variance by setting gamma-->infty, alpha-->0 and beta-->infty. The toy experiment in the paper has a densely distributed training points, thus the proposed method performs well. But I would like to see another experiment with sparsely distributed training points, in which case we can see better on the overfitting issue. 2, Scaling to high dimensional datasets. The proposed locality-aware mini-batching relies on a reliable distance measure, while the paper uses Euclidean distance. However, for high dimensional datasets, the Euclidean distance is hardly to be trusted. Similarly for the inducing point in variance network, the distance selection has the same issue. However I feel this this is not fatal, as RBF is known to work well, and other metric learning methods can be applied. 3, Computational complexity. Although the locality-aware mini-batching supports stochastic training, searching for the k-nearest-neighbour takes O(k N^2) computational cost, which might not be feasible in large datasets. In particular, if you learns a metric (Check Q3) along training, the searching process needs to be repeated, which makes it unfeasible. 4, I am not convinced by the local-likelihood analysis for location-aware mini-batching. Because the location-aware mini-batching is also an unbiased estimation for the total sum of log likelihoods, at end of the day, it should converge to the same point as standard mini-batching (of course there might be optimization issues). 5, I think Eq5 is a typo, that it should be sigma^2 (1- v) + eta * v Strengths, I have covered it in the contribution part.

[Author Response · NeurIPS 2019]

## NeurIPS Rebuttal for "Reliable training and estimation of variance networks"

We thank the reviewers for their constructive and fair reviews. We here address the key concerns, and note that the paper will be updated accordingly. We will discuss two shared concerns, and then move to individual reviewers.

First, however, we emphasize that the paper provides the first study of training issues related to variance networks, and develops new training methodologies that result in significant gains over current state of the art.

**High dimensionality (R2+R3):** Two reviewers expresses concern about extending our methodologies to higher dimensions. This is reasonable as both the locality sampler and the variance extrapolation depend on Euclidean distances, which are known to perform poorly in high dimensions. While we acknowledge the issue, we want bring additional perspectives to the discussion:

- One definition of variance is that it is the mean squared deviation from the mean, so the notion of variance itself is dependent on the Euclidean distance. This may indicate that variance networks are inherently difficult in high dimensions, where some regularity assumptions may be needed.
- While Euclidean distances behave poorly in high dimensions, there is plenty of empirical evidence that nearest neighbors are somewhat better behaved. As an example, spectral learning techniques have been very successful in moderately high dimensions. This is also in line with our observation that the proposed methods work well on MNIST, which is 768 dimensional (medium-sized data).
- As R3 mentions, the issue can possibly be circumvented through a different choice of metric; possibly one that is learned. This is an interesting venue for further research.

**Computational costs (R2+R3):** We agree that performing nearest neighbor search in the inner training loop presents a scalability issue. Two practical comments are, however, in order:

- We can rely on fast approximate nearest neighbor algorithms. This is an established area of research, where, in particular, randomized algorithms are currently showing notable speed gains.
- The variance network is typically trained on a GPU while the nearest neighbor search is performed on the CPU. Thus, the two processes can in principle be performed in parallel. The total training time can then be unaffected as long as the neighbor search is faster than a forward and a backward pass.

That being said, we acknowledge the concern, and view this as a topic for future work.

**We consider** the above two concerns to be both valid and important, but we also maintain that the paper provides the first working implementation of variance network training, which is a significant contribution.

**R1:** We agree with the reviewer that the three variance problems discussed in this paper (underestimation, trivialization, extrapolation) could easily each be the subject of its own paper. With this paper we aim at bringing attention to the multiple separate problems within uncertainty estimation, and argue that each problem requires its own solution. We provide initial solutions that significantly improve on the current state of affairs, but do expect that they can all be improved. We, further, would not be surprised if different application areas might benefit from slightly different solutions. We will update the paper with a discussion of these matters in more detail.

**R3:** We agree that the proposed methodologies do not solve the problem of overfitting. Interestingly, this problem also depends on the flexibility of the mean function. If this perfectly fits the data, then overfitting of the variance becomes more likely. This indicates that the link between variance estimation and regularization of the mean, could use further study; we consider this to be an interesting venue for research.

In practice we do not believe that the bias introduced by the sampling scheme plays much of a role, if the data is dense. However, theoretically we could have regions of data that are under-trained because they are underrepresented in the samples. The reason for the improved result is more likely due to an optimization that avoids the local minima of the usual mini-batching.

Regarding, Eq. 5: Yes, this is a typo. Thanks for catching it!

[Meta-Review · NeurIPS 2019]

The authors identify problems with estimating predictive variance using neural networks, and propose solutions to fix them. All the reviewers agreed that the paper is well-written, clearly highlighting the limitations of current methods and demonstrating that the proposed solution works better. The reviewers gave some suggestions to improve the paper, and raised some questions about computational complexity and scalability to high dimensions. I encourage the authors to take these into account when they prepare the final version.